# Bridging semantics and pragmatics
# in information-theoretic emergent communication

**Eleonora Gualdoni**
Apple MLR*
Universitat Pompeu Fabra
e_gualdoni@apple.com

**Mycal Tucker**
MIT
mycal@mit.edu

**Roger P. Levy**
MIT
rplevy@mit.edu

**Noga Zaslavsky**
NYU
nogaz@nyu.edu

## Abstract

Human languages support both semantic categorization and local pragmatic inter-actions that require context-sensitive reasoning about meaning. While semantics and pragmatics are two fundamental aspects of language, they are typically studied independently and their co-evolution is largely under-explored. Here, we aim to bridge this gap by studying how a shared lexicon may emerge from local pragmatic interactions. To this end, we extend a recent information-theoretic framework for emergent communication in artificial agents, which integrates utility maximiza-tion, associated with pragmatics, with general communicative constraints that are believed to shape human semantic systems. Specifically, we show how to adapt this framework to train agents via unsupervised pragmatic interactions, and then evaluate their emergent lexical semantics. We test this approach in a rich visual domain of naturalistic images, and find that key human-like properties of the lex-icon emerge when agents are guided by both context-specific utility and general communicative pressures, suggesting that both aspects are crucial for understanding how language may evolve in humans and in artificial agents.

## 1 Introduction

Languages evolve through repeated interactions in rich contexts, where various communicative and non-communicative goals co-exist. The conveyed meaning is often shaped by the local conversational context of utterances (Figure 1), as captured by the *pragmatic* behavior of interlocutors [1]. For example, the word 'player' can be interpreted as a baseball batter, catcher, or a guitar player, depending on the conversational context that shapes the listener's beliefs about the speaker's state of mind. That is, understanding meaning in context requires pragmatic reasoning about other agents' intentions and beliefs [1–3]. At the same time, words are associated with non-contextualized meanings, as captured by *lexical semantics*. For example, we have a shared idea of what 'player' means, regardless of any specific context in which this word might appear, and we can use it to communicate with new conversational partners in new contexts. While semantics and pragmatics are widely studied, their interface and co-evolution is largely under-explored and not well understood. In this work, we focus on a key open question at the interface of lexical semantics and pragmatic reasoning: How can a shared human-like lexicon emerge from local context-sensitive pragmatic interactions?

To address this question, we build on a framework for information-theoretic emergent communication in multi-agent reinforcement learning systems [4]. This framework is particularly relevant for addressing our question because it integrates task-specific utility maximization — a central component in well-established models of human pragmatic reasoning [2, 3, 5], which has also been center-stage in the literature on emergent communication in artificial agents [6–8] — with general information-theoretic constraints that are believed to shape human lexical semantic systems [9–13] and have

---

*Work finished prior to joining Apple

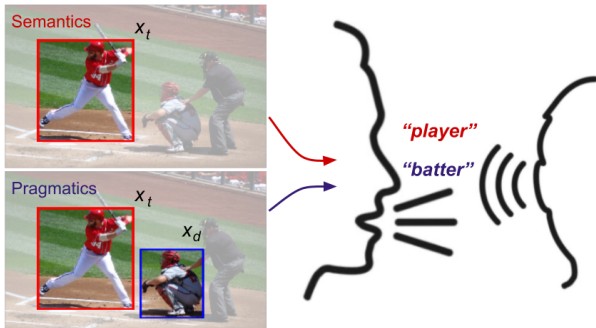

Figure 1: An example of an image from the ManyNames dataset illustrating both semantic and pragmatic communication. **Top** (semantic setting): A speaker communicates a target object $x_t$ (red box) by naming it regardless of the context induced by the image. The ManyNames dataset includes such responses from native English speakers (in this example, 10 speakers produced 'man', 10 'batter', 5 'baseball player', 4 'player', and 3 'person'). **Bottom** (pragmatic setting): A speaker and a listener observe two objects in an image (red and blue boxes). Only the speaker knows which one is the target $x_t$ and which one is the distractor $x_d$. The speaker's goal is to communicate $x_t$ given this shared context, and the listener's goal is to discriminate the target from the distractor.

recently also been applied to human pragmatic reasoning [14]. These information-theoretic constraints are derived from rate-distortion theory [15, 16], and more specifically, the information bottleneck (IB) principle [17], characterizing semantic systems in terms of efficient compression of meanings into words by optimizing the IB tradeoff between the complexity and informativeness of the lexicon [9]. Tucker et al. [4] developed a framework for training agents in emergent communication settings by integrating utility maximization with the IB objective, yielding a multi-objective optimization problem that trades off maximizing task-specific utilities with maximizing task-agnostic communicative informativeness and minimizing communicative complexity.

Here, we extend that framework to explicitly model the co-evolution of semantics and pragmatics. In our setup, artificial neural agents learn to communicate in a pragmatic setting, i.e., in the presence of a shared conversational context that may alter the meaning of their communication signals (illustrated intuitively in the bottom image of Figure 1). Importantly, agents are trained via self-play, without any supervision or exposure to human languages. At test time, we assess the pragmatic competence (i.e., utility) of the agents as well as the 'human-likeness' of the shared emergent lexicon that they converged on, by invoking it in a lexical semantic setting (illustrated intuitively in the top image of Figure 1, and see Figure 2 for a full description of our model's architecture). Given the substantial empirical evidence that human semantic systems are pressured to optimize the IB tradeoff, we predict that a shared human-like lexicon will not emerge when agents are guided solely by utility maximization but rather when they are guided by a non-trivial tradeoff between optimizing utility, informativeness, and complexity. Our goal in this work is to test this prediction, and more generally, to characterize the landscape of emergent communication systems induced by our novel information-theoretic framework for the co-evolution of semantics and pragmatics.

To this end, we consider a rich visual domain of naturalistic images provided by the ManyNames dataset [18], which also contains free-naming human data generated by native English speakers. This domain allows us to train agents across many different conversational contexts and then evaluate their emergent lexicon with respect to the English naming data. In support of our prediction, we find that human-like properties of the lexicon (its size, complexity, and alignment with English speakers), together with high pragmatic competence, emerge when agents are guided by both context-specific utility and general communicative pressures as derived from the IB principle. Interestingly, pressure for communicative informativeness, rather than (non-communicative) task utility, appears as the main driver of emergent communication, but weaker pressures to minimize complexity and maximize utility are still crucial for achieving human-like properties of the lexicon.

Our work is significant both from a cognitive science perspective and from a machine learning perspective. From a cognitive science perspective, our work provides a novel computational framework for studying the under-explored interface between lexical semantics and pragmatic reasoning, and our findings suggest that human languages may evolve under pressure to optimize a tradeoff

between task-specific utilities and general communicative constraints. From a machine learning perspective, our work demonstrates how cognitively-motivated optimization principles, implemented in neural network agents as intrinsic training objectives, can facilitate the emergence of interpretable human-like communication systems without any human supervision.

## 2 Related work

Our work integrates cognitively-motivated information-theoretic principles that are believed to underlie human language evolution, with deep learning tools for studying emergent communication in artificial agents, in order to develop a computational account of the co-evolution of lexical semantics and pragmatic reasoning. Semantics and pragmatics constitute two subfield in cognitive linguistics. While both focus on meaning in language, they capture largely complementary aspects of meaning. Lexical semantics is generally concerned with word meanings [19, 20], independent of any specific conversational context, whereas pragmatics is concerned with language use in context, typically assuming a known shared lexicon [1, 21, 3]. Our work focuses on the underexplored interface between these two aspects of language, departing from the traditional assumption that the lexicon is given a-priori and shared among pragmatic interlocutors.

Contemporary cognitive approaches to lexical semantics argue that word meanings are shaped by pressure for efficient communication [22, 23, 10]. Most relevant to our work, is the information-theoretic framework for semantic systems, proposed by Zaslavsky et al. [9], that predicts that human semantic systems evolved to optimize the information bottleneck (IB) trade-off [17] between the complexity of the lexicon (roughly, how many bits are allocated for communication) and its informativeness (roughly, how well a listener can understand a speaker, regardless of context). This framework has shown to account for the structure of semantic systems across hundreds of languages and multiple domains [9, 24, 11, 12], as well as semantic change over time [13]. Here, we leverage this empirically-supported theoretical framework to guide our interactive agents.

Research on pragmatic communication focuses on how speakers' lexical choices and listener's interpretations are affected by their local conversational context [1, 3, 25, 26]. Prominent models of pragmatics, such as the Rational Speech Act (RSA) framework [3], are grounded in game theory (see also [2]). In this view, pragmatic behavior is formulated within a cooperative reference game, where agents pragmatically reason about each other's communicative intentions with the goal of maximizing the game's utility. While these models enjoy broad empirical support [3], they assume that the underlying lexicon is given and shared across speakers and listeners, even when applied in reinforcement learning settings [27]. Recently, a theoretical link between the RSA framework and rate-distortion theory (RDT) has been derived [14], connecting pragmatic reasoning with general informational constraints that are closely related to the aforementioned IB trade-off. However, the RDT approach to pragmatics has also assumed a given lexicon [14], and more generally, the implied information-theoretic link between semantics and pragmatics has not yet been explored. Our work explored this potential link by relaxing the assumption that the lexicon is given to our agents, and instead studying how they may develop on their own a near-optimal human-like lexicon via a training objective that takes into account informational constraints.

Relatedly, Brochhagen et al. [28] developed a game-theoretic model for the evolution of the division of labor between semantics and pragmatics, and tested the model in a relatively small domain. In comparison, we consider here different learning dynamics and objective function, employ state-of-the-art deep-learning tools for training agents at scale, and evaluate our approach with respect to actual human data in a rich domain of naturalistic images [18]. McDowell and Goodman [29] considered the role of pragmatics in learning lexical meanings in deep-learning agents. However, they trained their agents in a supervised manner with respect to human-generated pragmatic data. In contrast, we are interested in how pragmatics and lexical semantic may emerge without any human supervision.

The emergent communication literature focuses on how agents may learn to communicate in reinforcement learning settings, without exposure to human-generated linguistic data [6, 7, 30]. While this body of literature largely focuses on utility (or reward) maximization, Chaabouni et al. [8] showed that utility-based emergent communication can lead to IB-efficient systems. In their settings, however, different complexity-informativeness tradeoffs can only be determined by external environmental factors. Tucker et al. [4] showed how to directly integrate the IB objective function with utility maximization in emergent communication and demonstrated the advantages of this framework for

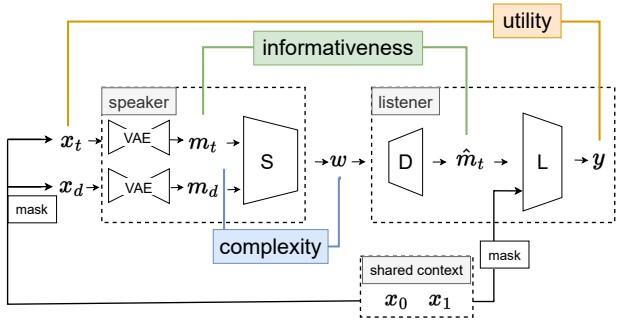

Figure 2: Communication model for the co-evolution of semantics and pragmatics (see Section 3). Agents are trained in a pragmatic setting, where both observe inputs $x_0, x_1$ as shared context; one input is randomly selected as target $x_t$ for the speaker. After training, the agent's emergent communication systems are evaluated in a lexical semantics setting, without shared context; the speaker observes only $x_t$ while $x_d$ is masked, and both inputs are masked for the listener.

faster convergence rates and better out-of-distribution generalization [31]. However, they did not study the co-emergence of semantics and pragmatics. Our work directly builds on the framework of Tucker et al. [4, 31] and extends it for studying the interface between semantics and pragmatics.

## 3 A unifying model for the co-evolution of semantics and pragmatics

The information-theoretic framework for emergent communication that we build on [4, 31] consists of an objective function that integrates utility maximization with the IB principle, and a communication model that specifies the agents' architectures. Our model (Figure 2) builds on Tucker et al. [4]'s proposed method, called vector-quantized variational information bottleneck (VQ-VIB), which we modify in order to be able to model the co-evolution of semantic and pragmatics; that is, to be able to train agents in a pragmatic setting and evaluate their communication, at test time, in both pragmatic and lexical semantic settings. The initial VQ-VIB architecture includes a speaker and a listener tasked to jointly solve a reference game. In our case, the speaker is defined by (i) a pre-trained variational autoencoder [VAE, 32] that provides a visual representation module for mapping an input $x$ to a 'mental' representation $m$; and (ii) an encoder module $S$ that generates a communication signal $w$ given the speaker's mental state. The listener is defined by (i) a decoder $D$ that observes $w$ and generates a reconstruction $\hat{m}$, and (ii) a policy $L$ for solving a downstream task.

In our **pragmatic setting** both agents observe a shared context $(x_0, x_1)$, while the speaker also observes which referent is the target $t$ and which is a distractor $d$ (more details about this step are provided in Section 4). The speaker then aims to communicate the target $x_t$. The VAE representation model is applied to $x_t$ and $x_d$ independently, generating $m_t$, the speaker's mental representations for the target, and $m_d$, the speaker's mental representations for the distractor. The listener's task is to guess the target based on $y = L(\hat{m}, (x_0, x_1))$. We wish to emphasize that in contrast to the standard approach in pragmatic models, in our setup the agents are not given a shared lexicon but rather implicitly learn it through their local, context-sensitive pragmatic interactions.

Agents are trained by optimizing a tradeoff between maximizing expected utility, maximizing informativeness, and minimizing complexity. The utility term, $U(x_t, y)$, is task-specific, and here we take it to be the accuracy of the listener's downstream decisions. Informativeness and complexity are task-agnostic communicative objectives, derived from the Information Bottleneck (IB) framework for semantic systems [9]. In this framework, the speaker's and listener's mental representations, $m_t$ and $\hat{m}$, are defined as belief states (i.e., probability distributions) over features in environment. Informativeness is related to the distortion between $m_t$ and $\hat{m}$, defined by the Kullback-Leibler (KL) divergence $D[m_t \| \hat{m}]$, such that low expected distortion amounts to high informativeness. Complexity corresponds to the mutual information between speaker's meanings and words, $I(m_t; w)$, which is roughly the number of bits that the agents will need for communication. Following [4, 31], we optimize a bound on the IB terms for practical reasons, as direct optimization of those terms does not scale. For informativeness, we treat $m_t$ and $\hat{m}$ as the means of the agents' mental distributions and then approximate informativeness by their MSE. For complexity, we consider a known variational

bound on the mutual information, denoted here for simplicty by $\tilde{I}$ [for more details and full derivation of the objective function, see 4]. Overall, the training objective is to maximize

$$\lambda_U \mathbb{E}\left[U\left(x_t, y\right)\right] - \lambda_I \mathbb{E}\left[\|m_t - \hat{m}_t\|^2\right] - \lambda_C \tilde{I}(m_t, f(I); w), \tag{1}$$

where the $\lambda$s are non-negative tradeoff weights that sum to 1, and $f(I)$ can be seen as a function that extracts two objects (a target and a distractor) from an image.

Optimizing only the utility term in the objective function, without any pressure for alignment between the speaker's and listener's representations (i.e., high informativeness), is expected to lead to lexical systems that are biased towards success in the specific training downstream task, and such systems are likely to depart from human lexical systems. On the other hand, maximizing informativeness alone will lead to highly complex and task-agnostic systems, and minimizing complexity alone will lead to no communication. None of these extremes seems human-like, and we therefore expect that human-like systems will emerge when all tradeoff parameters are active (i.e., non-zero).

Our **semantic setting** is designed to evaluate the emergent lexicon at test time, lending a window into how the agents use their words irrespectively of any local context. In this setting, only the target is shown to the speaker (the distractor is masked) and then the listener reconstructs $\hat{m}_t$ based on the speaker's word $w$, without any additional context or downstream task. This task resembles in its nature the task with which annotations for ManyNames –our dataset of interest [18], see below– are gathered, i.e. a naming task where annotators were asked to freely produce names for target objects appearing in natural images, identified with bounding boxed. Thus, we can use these naming annotations at test time to evaluate the agents' emergent semantics.

## 4 Experimental setup

### 4.1 The ManyNames domain

We train and test our agents on the ManyNames dataset [18], which contains 25K naturalistic images (see Figure 1 for example), each with one target object, appearing in a bounding box, annotated with $\sim 36$ names provided by English native speakers asked to freely produce a name (one word) to describe the object.[2] The name produced by the majority of the annotators for a target object is called the *topname*. In the case of Figure 1, top image, 'man' and 'batter' are equally probable topnames. Objects in the ManyNames dataset are also annotated according to their high-level semantic domain, which can be *people*, *animals-plants*, *buildings*, *vehicles*, *clothing*, *food*, or *home*. Importantly, our agents only observe the images in the dataset. They are not trained on any of the linguistic labels, which are used only for evaluation.

### 4.2 Data selection for pragmatic training

With respect to the choice of the target-distractor pairs for our pragmatic training setup (Figure 1, bottom image), our intuition is that identifying competing objects appearing in the same image, e.g., a batter and a baseball player in the same field, or a car and a taxi on the same street, instead of using random objects cropped from different images, should encourage the emergence of more natural semantic partitions, since agents will need to create lexical entries to solve naturally occurring ambiguities. For each image in the dataset, we consider the target object highlighted in the ManyNames annotation; and an additional object that we individuate through automatic object detection and a few filtering filtering steps. We used the Bottom-Up object detection model proposed by Anderson et al. [33], which incorporates a Faster R-CNN architecture [34] for object detection, and a ResNet-101 architecture [35] for feature extraction. Since Anderson et al. [33]'s model is fine-tuned on the VisualGenome image dataset [36], of which ManyNames images are a sample, we are guaranteed that this model can make good quality predictions on ManyNames. After running the object detector model on our images, we filter the detected bounding boxes keeping only those with Intersection over Union smaller than 0.1 with the ManyNames target, and that did not have size smaller than 10% of the target size.[3] We then computed the similarity between the candidates' visual features –automatically

---

[2]Creative Commons Attribution 4.0 International License.

[3]For $2.5K$ images, typically depicting only one large object as the ManyNames target, we do not find any detected object with the desired properties. Being unable to use these images in the pragmatic setting, we exclude them from our data sample.

| Model | $\Delta$Compl. | $\Delta$LexSize | NID | Utility | MSE |
|---|---|---|---|---|---|
| $\lambda_U = 1$ | 1.76 ($\pm$0.31) | 559 ($\pm$382) | 0.84 ($\pm$0.02) | 0.95 ($\pm$0.00) | 20 ($\pm$7.18) |
| $\lambda_I = 1$ | 2.80 ($\pm$0.06) | 1687 ($\pm$107) | 0.60 ($\pm$0.00) | 0.83 ($\pm$0.00) | 0.13 ($\pm$0.00) |
| $\lambda_C = 1$ | 5.21 ($\pm$0.01) | 381 ($\pm$0.98) | 0.99 ($\pm$0.00) | 0.58 ($\pm$0.00) | 0.32 ($\pm$0.00) |
| $\lambda^*_{\text{Eng}}$ | **1.48** ($\pm$0.08) | **22** ($\pm$35) | **0.55** ($\pm$0.00) | 0.72 ($\pm$0.01) | 0.23 ($\pm$0.00) |

Table 1: Evaluation of the model that is best aligned with English, $\lambda^*_{\text{Eng}}$ ($\lambda_U = 0.005$, $\lambda_I = 0.98$, $\lambda_C = 0.015$), in comparison with three baselines: $\lambda_U = 1$, corresponding to utility maximization without any informational constraints, which is the most common objective function in the emergent communication literature; and $\lambda_I = 1$ and $\lambda_C = 1$ which correspond to the other two extremes of either maximizing informativeness or minimizing complexity. $\Delta$Compl. and $\Delta$LexSize are the absolute value of the differences between the complexity or lexicon size of the emergent system and English. NID measures the misalignment between the emergent system and English. For all three measures, lower values reflect a better fit to the human data. Utility and MSE correspond to the agent's pragmatic competence and reconstruction error, respectively. Each cell shows the mean value across three random seeds $\pm$ SEM.

extracted by the ResNet-101 incorporated in Anderson et al. [33]'s model– and the ManyNames target's features, choosing the detected object with the highest similarity value. This final selection step based on visual similarity aims at providing our agents with some challenging cases to solve, like the one shown in Figure 1, where communication needs to allow the discrimination between two objects from the same semantic category, e.g. a batter and another baseball player.

At the end of this procedure, we obtained two objects per image: the ManyNames target; and the additional object detected by us. We used these two objects to train our agents, choosing our target's identity randomly, making sure that in 50% of the cases the target would be the larger object, and in the other 50% the distractor would be larger.

### 4.3 Multi-agent simulations

We trained 270 agent pairs with a range of combinations of $\lambda_U$, $\lambda_C$ and $\lambda_I$. We used parameter annealing which, besides being a computationally efficient alternative to training agents from random initializations, has been shown to capture at least some aspects of the evolution of human semantic systems [9]. We used a pretrained ResNet18 [35] to extract 512 dimensional features from target and distractor objects before passing them to the agents.[4] In all our experiments, we trained agents on 70% of the images (randomly sampled) in self-play, with no human supervision, batch size of 128, and codebook initialized with 3000 trainable communication vectors (see Appendix A for further details). [5]

## 5 Results

### 5.1 Quantitative evaluation

We report quantitative results for the 5 evaluation metrics shown in Figure 3, for each set of $\lambda$s, at test time. The top panel shows the three components of the agents' training objective: (a) the agents' pragmatic competence, as measured by their test-time expected utility on unseen images, in the pragmatic setting; (b) the reconstruction loss between the speaker's and listener's mental representations, measured by the negative MSE (values closer to 0 corresponds to higher informativeness); and (c) the complexity of the emergent lexicon. The bottom panel of Figure 3 shows two measures for assessing the human-likeness of the emergent systems: (d) the lexicon size of the artificial systems,

---

[4]We extract visual features with a ResNet18 model, that is a standard choice for object classification, and choose not use the visual features from Anderson et al. [33]'s model, which we employ for the object detection phase. This is because this model, being specialized for object detection, produces visual features for objects in bounding boxes that carry some degree of information about the image context. This is a desirable property when performing object detection, but not for our experiments.

[5]Our code is available at `https://github.com/InfoCogLab/info-sem-prag-neurips2024`.

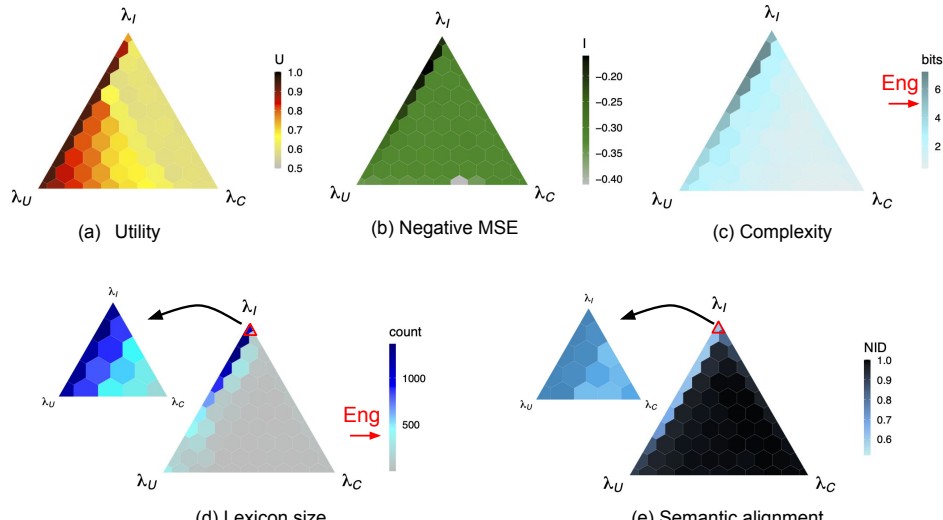

Figure 3: **The information-theoretic landscape of emergent communication**. Each simplex shows a test-time evaluation metric using either pragmatic (a) or semantic (b-e) settings, for the range of models spanned by the values of $\lambda = (\lambda_U, \lambda_I, \lambda_C)$. Overall, none of the extremes yield human-like communication, and the best alignment with English is achieved for $\lambda > 0$ near the top of the simplex. **(a)** Utility, reflecting the agents' pragmatic competence in solving the discrimination task. **(b)** Reconstruction loss, measured by the negative MSE between the speaker's and listener's target representations; values closer to $0$ correspond to more informative communication (for ease of visualization, the model trained with $\lambda_U = 1$, with negative MSE $-20$, is ignored in this plot). **(c)** Complexity, measured by the mutual information between the speaker's inputs and communication signals. The red arrow indicates the complexity of English as estimated from the ManyNames dataset. **(d)** The effective lexicon size of the emergent systems (i.e., the number of signals that are used with non-zero probabilities). The red arrow indicates the number of unique English terms in the ManyNames dataset. **(e)** Semantic (mis)alignment between the emergent systems and English, measure by Normalized Information Distance (NID). Lower values correspond to better alignment. **(d-e) Insets** show the top of the simplex with higher resolution, revealing the importance of $\lambda_C > 0$ in attaining human-like lexicon sizes and better alignment with English.

in comparison to the number of English words used in the ManyNames dataset; and (e) the semantic (mis)alignment between the emergent systems and the English naming system, measured by the Normalized Information Distance [NID 37]. NID takes into account for full meaning-to-form mappings and is bounded between $[0, 1]$, with lower values corresponding to better alignment. The measures in (b-e) are evaluated in the semantic setting, which was not used during training, across the full dataset.

As expected, agents trained with large weights on one of the three $\lambda$s develop communication systems skewed towards one component of the objective, yielding non-human like solutions at the extremes of the $\lambda$ simplex. Next, we characterize the landscape of the emergent systems and discuss the human-like tradeoffs that best capture the linguistic behavior of English speakers.

**High $\lambda_U$ regime.** High values of $\lambda_U$ (bottom left corner) allow agents to achieve very high pragmatic competence (utility $> 0.9$, Figure 3a), maximizing the listener's success in solving the downstream task (i.e., distinguishing the target from the distractor), but result in very poor alignment between the emergent lexicon and the English lexicon (NID values $> 0.8$, Figure 3e). This finding resonates well with the cognitive science literature on the role of informational constraints in the evolution of human semantic systems [10], as well as with empirical findings that suggest that humans do not achieve maximal utility in our task when restricted to using only lexicalized items [26].[6] In other words, in this regime, which focuses primarily on utility maximization without information

---

[6]In our settings, agents can communicate only using lexicalized items. The usage of more complex structures, such as syntactic constructions, is beyond the scope of the present work and is left for future research.

constraints, agents are likely to develop communication strategies that do not align with human-like lexical semantics as they are pressured to over-compensate for the lack of syntactic constructions.

**High $\lambda_I$ regime.** High values of $\lambda_I$ (top corner) lead to highly informative communication (negative MSE $> -0.20$, Figure 3b). However, as expected, this comes at the cost of large lexicon sizes and high complexity. Agents trained with $\lambda_I$ values close to 1 learn to use thousands of words for the ManyNames domain ($> 1.5K$ vs. $\sim 400$ topnames used by English speakers for $\lambda_I = 1$, Figure 3d), and too complex communication systems ($> 6$ bits vs. $\sim 5$ bits in English, Figure 3c). Compared to $\lambda_U$, high values of $\lambda_I$ seem to generally favour semantic alignment (NID $\sim 0.6$, Figure 3e), although it is important to note that the best NID is not achieved with $\lambda_I = 1$, but rather with a non-trivial combination of all tradeoffs, as discussed below. Finally, it is noteworthy that highly informative lexical systems also yield high utility (e.g., utility $> 0.9$ and negative MSE $> -0.18$ for $\lambda_U < 0.3$, $\lambda_I > 0.7$, $\lambda_C = 0.0$). This is in line with findings from Tucker et al. [38]. Indeed, in our framework, utility and informativeness are not in complete competition, but rather capture complementary aspects of successful communication that are only partially aligned.

**High $\lambda_C$ regime.** High values of $\lambda_C$ (bottom right corner) encourage minimal complexity and lead to unsuccessful communication, both in terms of utility and informativeness, as well as to very small lexicon sizes and low semantic alignment. In general, $\lambda_C$ seems to act at a very small scale, with important effects on the lexicon already perceivable at very small values (see, for instance, the large decrease in lexicon size for $\lambda_C \sim 0.02$ in Figure 3d, as well as the corresponding increase in semantic alignment in Figure 3e).

**Human-like tradeoffs ($\lambda_{\text{Eng}}^* > 0$).** A combination of pressures fosters the emergence of natural solutions. Table 1 summarizes key properties of the landscape that we have explored. As expected, the three extremes on the simplex lead to unnatural, non-human-like solutions. Moving away from the extremes, we identify a non-trivial tradeoff, $\lambda = (\lambda_U = 0.005, \lambda_I = 0.98, \lambda_C = 0.015)$, that achieves the best fit with respect to the English data. We thus refer to this model as $\lambda_{\text{Eng}}^*$. This model achieves the best semantic alignment with English (Figure 3e), and roughly matches the English lexicon size (Figure 3d) and complexity rate (Figure 3c). It also achieves good reconstruction (Figure 3b) and high pragmatic competence (Figure 3a).

These quantitative findings support our predictions and demonstrate how our framework can advance our understanding of the co-evolution of semantics and pragmatics. To further understand the communication strategies learned by our agents, we next turn to a qualitative exploration of the emergent communication systems.

## 5.2 Qualitative evaluation

Figure 4 offers a visualization of the agents' communication. We plot as dots the visual features (reduced to 2D via PCA) of 500 objects randomly sampled from 3 categories in ManyNames ('woman', 'giraffe', and 'train', i.e. the most frequent names in the semantic domains of 'people', 'animals/plants', and 'vehicles'), identified by color. White crosses correspond to the listeners' reconstructions ($\hat{m}_t$, see Figure 2) in the semantic setting, which roughly represent word meanings.

Figure 4a, b, and c illustrate the solutions learnt by the models at the edges of our simplex. When trained with $\lambda_U = 1$ (Fig. 4a), agents are only driven by the task-related utility, i.e., maximizing success in pragmatic interactions. In this scenario, the listener does not learn to reconstruct a mental representation of the object, and the solution lacks a robust, non-contextual semantics. When trained with $\lambda_I = 1$ (Fig. 4b), in order to maximize informativeness, agents learn highly complex solutions, with large lexicons mapping words to small sets of objects, and not identifying human-like categories. When trained with $\lambda_C = 1$ (Fig. 4c), agents compress their lexicons at the cost of losing important distinctions, and achieve a solution where the same word describes all the objects. This solution does not enable successful communication.

In contrast, our $\lambda_{\text{Eng}}^*$ model (Fig. 4d), trained with a tradeoff between utility, complexity, and informativeness, starts approaching a natural solution, learning word meanings that roughly map to the human categories. This solution is simple, yet it allows for informative communication and successful pragmatic interactions. Still, the agents seem to have learnt additional words, capturing spurious distinctions, especially for the peripheral areas of each category cluster, and for the category

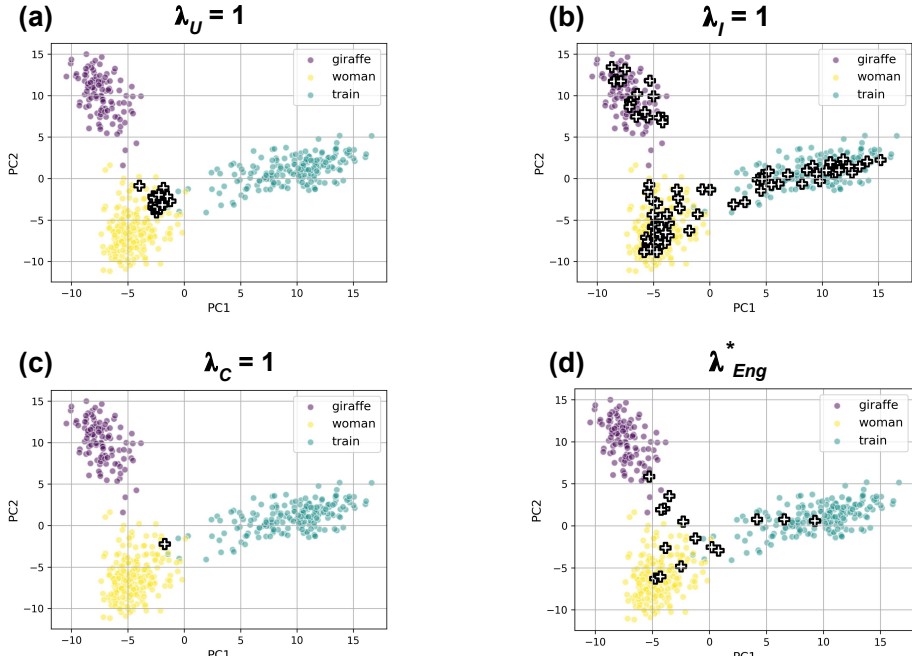

Figure 4: Visualization of the emergent communication systems for different combinations of $\lambda$s (best seed for each). In each plot, the colored dots correspond to a 2D PCA of the input features of $500$ randomly sampled objects from 3 distinct high-frequency categories in the ManyNames dataset ('woman', 'train', 'giraffe'). The white crosses correspond to the listener's reconstructions ($\hat{m}_t$, see Figure 2) given the speaker's communication signal, which roughly captures the meaning of each signal. **(a)** When agents are trained only with respect to utility, the listener does not learn to reconstruct a mental representation of the object. **(b)** When agents are trained only to maximize informativeness, the emergent communication system is highly complex, essentially assigning unique signals to much of the input space. **(c)** When agents are trained only to minimize complexity, a trivial non-informative system emerges that employs only one signal. **(d)** When agents are trained with respect to a non-trivial tradeoff between utility, complexity, and accuracy, a more human-like communication system emerge. These systems are as simple as possible while maintaining sufficient utility and informativeness.

'woman' (the most frequent one in ManyNames). Besides reasons related to the data distribution, our hypothesis is that the agents may have developed human-like categorizations for prototypical members of the human categories (i.e. those with visual features near the center of the category cluster), but also additional, non human-like categories for atypical objects. Indeed, atypical objects are harder to categorize for humans as well, and may not clearly belong to one single semantic category [39, 40]. This hypothesis may also explain why our $\lambda^*_{\mathrm{Eng}}$ model has an NID score of $0.55$, showing moderate, but not perfect, alignment with human semantics. We gather support for this explanation by computing the NID score only with objects prototypical for their category: on this set of objects, the NID for $\lambda^*_{\mathrm{Eng}}$ decreases to $0.45$ ($\pm 0.02$), suggesting higher alignment –see Appendix B for further details.

## 6    Conclusion

Words in the human lexicon are associated with non-contextual meanings, as well as shaped by the local conversational context. In this work, we have addressed a key open question for language evolution: How can a shared lexicon emerge from local context-sensitive interactions? We modeled the semantics-pragmatics interface by building on a framework for information-theoretic emergent communication in neural agents. We trained agents to interact in self-play in the presence of a shared

conversational context, guiding them with combinations of cognitively-motivated pressures. We then tested their pragmatic competence, as well as the human-likeliness of their emergent semantics. By exploring the landscape of emerging artificial languages, we demonstrate that, if trained with pressures for both context-specific utility and general communicative constraints, agents learn systems with key human-like properties and that allow for successful pragmatic interactions. Our findings inform current theories of language evolution, and show that cognitively-motivated optimization principles can facilitate the emergence of human-like communication strategies in neural networks.

## 7 Limitations

Our work aims to better understand the computational principles that underlie language evolution, in humans and artificial agents, with focus on the interface between semantics and pragmatics. While we presented an important first step towards this goal, we were only able to evaluate our model on English data and focused on the lexicon only. An important direction for future work is the evaluation of our model on a larger, more diverse set of languages. In addition, our work has focused only on the use of lexical items in communication. Therefore, another important direction for future research is to extend our framework to more complex communication structures, such as syntax, morphology, and compositional meaning.

One potential concern about our model is that it employs a pre-trained object classification model [35] to extract visual features. This pre-trained model was trained with classification labels. Thus, one might worry that our agents were implicitly exposed to some linguistic knowledge. We have several reasons to believe that this exposure is negligible. First, the classification labels are coarse while we evaluate the agents with respect to fine-grained naming data. Second, our agents are trained in a pragmatic setting, whereas the classification labels are non-contextual. Third, the majority of our agents develop non-human-like lexical systems, suggesting that the pretrained vision component is not sufficient for alignemnt with English. Having said that, in further work we intend to explore the influence of other types of visual features.

Finally, the NID score achieved by our agents, while encouraging, suggests that even our best-performing agents are not yet fully aligned with (English speaking) humans. Therefore, further research is needed to understand how to close this gap and guide our agents toward more human-like communication systems.

## Acknowledgements

This research was partially by supported by grant PID2020-112602GB-I00/MICIN/ AEI/10.13039/501100011033 from the Ministerio de Ciencia e Innovación and the Agencia Estatal de Investigación (Spain) and the European Union's Horizon 2020 research and innovation programme (grant agreement No. 715154). We thank the COLT group from Universitat Pompeu Fabra for feedback on this work.

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

## A  Details about agents' training

We started by training our agents with $\lambda_U = 1$ until convergence, i.e. for $10K$ epochs. After that, we first annealed models by keeping $\lambda_C$ fixed at 0, while gradually decreasing the value of $\lambda_U$ and increasing the value of $\lambda_I$, until reaching $\lambda_I = 1$. Then, for each trained value of $\lambda_U$, we gradually annealed $\lambda_C$. For each annealing step, we trained until reaching variance in the training objective lower than $0.0001$ for the latest $1K$ epochs (as a criterion for convergence). We followed a non-uniform annealing schedule, re-fined after an initial exploration phase, aimed at identifying regions in the simplex showing interesting patterns with respect to our metrics. In these regions, we sampled the combinations of parameters more densely, e.g. with steps of $0.002$, while in the other regions we sampled with steps of $0.1$. Agents were trained with batch size $128$, hyperparameter $\beta$ of the VQ architecture set at $1$. See Tucker et al. [38] for further details about the architecture and hyperparameters.

Experiments were run on a cluster with $12$ nodes with $5$ NVIDIA A30 GPUs and $48$ CPUs each. Training the $\lambda_U = 1$ model took around $30$ minutes. Training one annealed model could take up to $15$ minutes, often less. Computing evaluation metrics took a total of $10$ hours. We estimate the overall time required to run this analysis to be around $4$ days. Considering our exploration phase and failed experiments, we estimate the total runtime required by this paper to have been around $20$ days.

## B  Identifying prototypical objects

To identify in ManyNames objects that are prototypical members of their categories, we took the following approach: for each topname appearing at least 30 times in ManyNames, we selected the 15 most probable images based on the human annotations (in general, visual typicality for a name correlates with name probability [39, 40]). This process resulted in 69 words and a total of 1035 images. These images are, at worst, the $50\%$ most typical images for their human name. On this set, $\lambda^*_{\text{Eng}}$ achieves NID of $0.45$ ($\pm 0.02$).

