# OpenReview forum: "Bridging semantics and pragmatics in information-theoretic emergent communication"
_NeurIPS.cc/2024/Conference — NeurIPS 2024 poster_

### Official Review · Reviewer_CiKA · 2024-07-11

**Soundness:** 3
**Presentation:** 3
**Contribution:** 3
**Rating:** 7
**Confidence:** 3

**Summary:**

This study explores how language, combining meaning and context, might evolve. It examines how a shared vocabulary can emerge from interactions that consider the situation.  By training agents in an unsupervised fashion to consider both context-specific utility and general communication pressures, the research reveals these aspects are key for understanding language evolvement.

**Strengths:**

- Pragmatics is under-explored in NLP, but it is an important topic to study and model language. This work is an novel and interesting effort towards that direction.
- The approach seems reasonable, and the evaluation shows the potential of using cognitively-inspired optimization principles for the communication strategies akin to human interaction.

**Weaknesses:**

- I don't see major weaknesses.

**Questions:**

-  The paper's approach is interesting.  To just confirm my understanding: in this specific setting, pragmatic representations will be always more specific (like hyponyms) than semantic representations, rather than dealing with connotations. This seems similar to the challenge of finding the most accurate hyponyms to describe something. Is my understanding correct?

- What do you think would be the potential real-world applications of this work?

**Limitations:**

The paper addressed limitations.

---

> ### Author Rebuttal · Authors · 2024-08-07
>
> We thank the reviewer for the positive and encouraging feedback!
>
> To address the reviewer’s questions:
>
> > The paper's approach is interesting. To just confirm my understanding: in this specific setting, pragmatic representations will be always more specific (like hyponyms) than semantic representations, rather than dealing with connotations. This seems similar to the challenge of finding the most accurate hyponyms to describe something. Is my understanding correct?
>
> This is partially correct. While agents could learn the strategy described by the reviewer, they may also learn a mix of strategies that humans employ. For example, a related phenomenon is known as scalar implicatures, in which a weaker characterization, such as ‘dog’, may implicitly rule out references that correspond to stronger characterizations, such as ‘Pomeranian’ (imagine a scene with a Labrador and a Pomeranian; both are types of dogs but Labrador is considered a more typical dog compared to Pomeranian). Humans also employ strategies with non-hierarchical alternatives (Silberer et al., 2020). That is, different conceptualizations for the same object can be produced by speakers to highlight different aspects of the referent. For example, we can call someone “teacher” irrespectively of any context, but also refer to them as “tennis player” when the context requires us to do so, even though “teacher” and “tennis player” are not in a hypernym-hyponym relationship. Our agents may also converge on solutions of this kind.
>
> > “What do you think would be the potential real-world applications of this work?”
>
>
> While our work focuses on the science of intelligence, rather than on engineering AI applications, we see at least one important direction in which our work could potentially lead to improvement in real-world applications. Recall that our agents are trained without any human-generated linguistic data. This stands in sharp contrast to LLMs that require massive amounts of training data, which is unavailable in low-resource languages. Therefore, our work could help understand how to leverage cognitively-motivated optimization principles for training more data-efficient language models, with the hope of advancing more equitable language technologies. We thank the reviewer for raising this question and we will address this point in the paper.
>
>
> **References**
>
> Silberer, C., Zarrieß, S., and Boleda, G. (2020). Object naming in language and vision: A survey and a new dataset. In *Proceedings of the 12th Language Resources and Evaluation Conference*, pages 5792–5801, Marseille, France. European Language Resources Association.

---

> > ### Comment · Reviewer_CiKA · 2024-08-09
> >
> > Thank you for the response.

---

### Official Review · Reviewer_F8cg · 2024-07-12

**Soundness:** 3
**Presentation:** 4
**Contribution:** 3
**Rating:** 5
**Confidence:** 2

**Summary:**

This paper addresses the co-evolution of pragmatics, which must be interpreted according to context, and lexical semantics, the common meaning independent of context. In this paper neural agents are trained in the pragmatic setting to select the target object from two objects, and evaluated in the semantic setting to reconstruct mental representation of the target.

The authors used the vector quantize variation information bottleneck (VQ-VIB) to evaluate the trade-off of utility, informativeness, and complexity. The results showed that human-like outcomes emerged when not solely pursuing one aspect among utility, informativeness, and complexity, but rather making a non-trivial trade-off among them.

**Strengths:**

The idea of learning emergent communication in a pragmatic setting and evaluating it in a semantic setting is intriguing and meaningful.

This paper effectively conducts experiments on naturalistic images. Through quantitative and qualitative evaluation, the authors show that human-like lexicon emerges when optimizing both context-specific and general objectives.

**Weaknesses:**

The explanation for the significance of the scenario that training in a pragmatic setting and testing in a semantic setting is insufficient.
The paper's contributions are somewhat unclear. Except for the point that the communication model is trained in a pragmatic setting, it is similar to [1] and [2]. It uses VQ-VIB as presented in [1], and the overall communication model is similar to [2].

The novelty is somewhat unclear in the claim that optimizing both task-specific and general objectives yields human-like results. Although there is a difference in that emergent communication is learned in a pragmatic setting, the trade-off between task-specific utility and general information is already presented in [1].

[1] Tucker, Mycal, R. Levy, J. Shah, and Noga Zaslavsky. 2022. “Trading off Utility, Informativeness, and Complexity in Emergent Communication.” *Advances in Neural Information Processing Systems*.

[2] Tucker, Mycal, Roger P. Levy, Julie Shah, and Noga Zaslavsky. 2022. “Generalization and Translatability in Emergent Communication via Informational Constraints.” https://openreview.net/pdf?id=yf8suFtNZ5v.

**Questions:**

- What is the significance of training in a pragmatic setting and testing in a semantic setting?
- Since the training loss includes utility, informativeness, and complexity, isn't semantics considered during training?

---

> ### Author Rebuttal · Authors · 2024-08-07
>
> > The explanation for the significance of the scenario that training in a pragmatic setting and testing in a semantic setting is insufficient.
>
> > What is the significance of training in a pragmatic setting and testing in a semantic setting?
>
> We believe that this concern is somewhat related to the framing concern of reviewer Cm9N. As we explain in the related work section, and in our response to Cm9N, semantics and pragmatics are two key aspects of linguistic meaning that have so far been mostly studied separately from each other. While they are related, the interface between them has been largely under-explored and is not well understood. Most computational models of pragmatics assume a given lexicon, and most of the work of lexical semantics ignores the context-sensitive aspect of meaning that is captured by pragmatics. As we note in the abstract: “we aim to bridge this gap by studying how a shared lexicon may emerge from local pragmatic interactions.”
>
> We believe that clarifying the framing throughout the paper, as we explain in the global rebuttal, will address this concern.
>
> > The paper's contributions are somewhat unclear.
>
> Our contributions are stated in the abstract (lines 9-15), introduction (lines 68-75), and conclusion section (lines 294-304). We would appreciate it if the reviewer could explain what is unclear about these lines in the paper, so that we could address this concern. Otherwise, this comment by the reviewer is too vague.
>
> > Except for the point that the communication model is trained in a pragmatic setting, it is similar to [1] and [2]. It uses VQ-VIB as presented in [1], and the overall communication model is similar to [2].
>
> While our model builds on the work by Tucker et al. that the reviewer mentioned, we extend that work in several important ways. First, not only is our training procedure different, but also our communication model is different. To see this, compare Fig. 2 in our paper with Fig. 1 in each one of the papers the reviewer cites. The evident differences are crucial for our work: we adjust the model to incorporate a shared context for the speaker and listener and implement a masking mechanism that allows us to invoke the model in two different ways (pragmatics and semantics settings). Second, in contrast to Tucker et al., who considered a discrimination task between whole images, our task focuses on discrimination between co-occurring objects within single images. This is crucial in our case, as we aim to capture communication in naturalistic scenes. Finally, our work addresses a major open question that has not been addressed by Tucker et al. That is, we focus on the under-explored interface between semantics and pragmatics, while Tucker et al. have focused on establishing the utility-informativeness-complexity tradeoff, and on the generalization and translatability afforded by this new framework.
> Thanks to the reviewer’s comment, we have noticed that these important differences were not stated clearly enough in the related work section. We will adjust that section accordingly to address the reviewer’s point.
>
> > Since the training loss includes utility, informativeness, and complexity, isn't semantics considered during training?
>
> The key point in this context is that semantics is not built into the agents and we do not assume that a shared lexicon between the speaker and listener is given a-priori. This is in contrast to the vast majority of studies in computational pragmatics that assume a given lexicon (for review see: Goodman, & Frank, (2016). Pragmatic language interpretation as probabilistic inference. *Trends in Cognitive Sciences*). Semantics is considered during training only implicitly, in the sense that we incorporate in our training objective a general optimality principle, the information bottleneck, that is believed to guide the evolution of human semantic systems.

---

### Official Review · Reviewer_Cm9N · 2024-07-12

**Soundness:** 4
**Presentation:** 3
**Contribution:** 3
**Rating:** 7
**Confidence:** 4

**Summary:**

This paper looks at aligning emergent communication with human language through the joint optimization of the utility, informativeness, and complexity of a communication channel.
This is done in the context of a "pragmatic" signalling game where a speaker agent must refer to an object in an image contrasting it with a distractor object in the image.
The resulting emergent communication protocol is evaluated in terms of its lexical semantics such as lexicon complexity and size.

**Strengths:**

In conjunction with standard criteria, there are three characteristics that are particularly important for emergent communication research: reusability (how easily can another researcher use the products of this research), generalizability (how much do the findings of this research apply broadly to our knowledge of emergent communication), and directedness (does this research contribute concretely to particular questions in emergent communication research).

### Quality
- (major) The experiments are reasonably designed and demonstrate informative results.
### Clarity
- (major) The experiments are easy to follow and are accompanied with informative visualizations.
### Reusability
- Nothing of note.
### Generalizability
- (major) The UIC framework utilized in this work is a general way to think about emergent communication systems with support in psycholinguistic theories.  As a result, providing evidence of the utility of this framework as well as an example of how to use it makes such a framework more effective down the road for for emergent communication research.
### Directedness
- (major) As mentioned above, the UIC framework makes progress towards multiple goals within emergent communication research: developing more formal models to describe system behavior as well as aligning emergent language agents (and the languages themselves) with human languages behavior.

**Weaknesses:**

### Quality
- Nothing of note.
### Clarity
- (major) It is difficult to tell if the paper is primarily about relationship between semantics and pragmatics in a given EC system or about empirically verifying the utility-complexity-informativeness framework.
### Reusability
- (major) The code is not available to be reviewed.
- (major) Although the code is available for VQ-VIB, the details on how the code (or some other codebase) was adapted to this paper are sparse.
### Generalizability
- (minor) There are not many details on how the UCI framework could be applied to other emergent communication systems or questions.  I think the framework is already pretty general, so it would be too difficult to expand on this for a paragraph or two.
### Directedness
- Nothing of note.

**Questions:**

Although I think the contributions of this paper are worthwhile, I think it is somewhat caught between different framings.  From the introduction (and title), I got the sense that the paper was about semantics and pragmatics, but from the rest of the paper I got more the sense that the paper was about validating the informativity-complexity-utility tradeoff using emergent communication techniques with the ManyNames dataset.
I think if the framing of pragmatic/semantics is primary, then there need to be more explicit definitions of what semantics and pragmatics is, why the distinction is important, and what the empirical investigation is telling us about it; I understand the difference between the "pragmatic" and "semantic" goals in the environment, but I do not get an explicit sense of what the relationship is between these two and why it is important. Also, some of the framing seems to make pragmatics seem wholly independent of semantics when really pragmatics is more a pressure on semantics.
On the other hand, if the framing is to be more consistent with the empirical work as it stands currently, I think there ought to be more talk of why the IUC-tradeoff is important/useful both generally and specifically with relation to the experiments.
Addressing this issue adequately would likely lead to me raising my Rating/Presentation score.


### Comments
- Figure 2: "Agent are trained" -> "Agents are trained"
- Line 156: "in its nature to" -> "in it nature"
- Not a requirement at all but it would be informative to compare the EC results to the recently released Mandarin Chinese data in ManyNames (assuming the data is already in the right format for analysis).

**Limitations:**

N/A.

---

> ### Author Rebuttal · Authors · 2024-08-07
>
> **Clarity + Questions (major)**
>
> The main goal of the paper is to study the interface between semantics and pragmatics, and specifically, to address the question: how can a shared human-like lexicon emerge from task-specific, context-sensitive pragmatic interactions? As the reviewer noted in the “Questions” section of their review, this framing is clear in the title and introduction (we also believe it is clear in the abstract, related work, and conclusions), but less so in other parts of the paper. We thank the reviewer for noticing this, and believe that this shortcoming is due to our focus on the technical details in the more technical parts of the paper, such as the experimental setup and results.
>
> To address this issue, the reviewer suggested that “there need to be more explicit definitions of what semantics and pragmatics is, why the distinction is important, and what the empirical investigation is telling us about it”. We agree and are grateful for this suggestion. Here we address the reviewer’s questions, and in the global rebuttal we explain how we will revise the paper to address this concern.
>
> The term **lexical semantics** refers to the study of word meanings [1, 2], independently of any specific conversational context. An example we mention in the paper (lines 23-25) is that we have a shared idea of what the word ‘player’ means, regardless of any context or scene in which this word might appear. The structure and evolution of word meanings has been an active research area in linguistics and cognitive science for several decades [3-5], and it has recently been characterized using the information-bottleneck (IB) complexity-informativeness tradeoff  [6, 7]. However, the vast majority of studies in this area do not take into account how words are used in context, which can alter their meaning.
>
> The term **pragmatics** refers to the study of how the local conversational context may alter meaning in real-time, as speakers and listeners reason about each other’s intentions and beliefs [8, 9]. For example, a speaker may choose either ‘player’ or ‘batter’ to describe the target person in the red bounding box in Fig. 1. In scenes with only one player, the more frequent word ‘player’ may be easier to produce, but in scenes with more than one player this word would not be informative about the speaker’s intended meaning. Therefore, if the speaker takes into account how a rational listener would interpret its words, then in the scene shown in Fig. 1 the speaker should use the word ‘batter’ instead of ‘player’. This type of reasoning, called pragmatic reasoning, has been widely studied both experimentally and with computational models that typically use tools from Bayesian inference and game theory [e.g., 9-11], that emphasizes only utility maximization. However, the vast majority of studies in this area assume that the lexicon is given and fixed.
>
> The **distinction** between semantics and pragmatics is important for several reasons: First, they capture two different aspects of language: semantics captures how meaning is structured into words regardless of context, while pragmatics captures the context-sensitive meaning. Second, they capture two different time-scales: word meanings are stable building-blocks of language that change relatively slowly, while pragmatic reasoning applies in real-time, everyday language use. Finally, these two aspects of linguistic meaning constitute two subfields of linguistics, and therefore much of the relevant literature, including in computational linguistics and cognitive science, is organized around this distinction.
>
> At the same time, it is widely agreed that semantics and pragmatics are related, as the reviewer noted as well; however the interface between them has been largely underexplored. As noted above, these two aspects of language are typically studied independently (but see our related work section for a few exceptions). As we state in the abstract, **our computational and empirical investigation aims to bridge this gap in the literature**, by providing a unified computational framework for studying semantics and pragmatics simultaneously. To this end, we extend a framework that integrates the IB complexity-informativeness tradeoff that characterizes lexical semantics with utility maximization that has been used to model pragmatic reasoning. In addition, in our setup, we do not assume that the lexicon is given a-priori, in contrast to most models of pragmatics, but rather investigate how it may naturally emerge from context-sensitive interactions. Our results show that in order to understand how a human-like lexicon may emerge from pragmatic interaction, it is crucial to consider all three terms in our objective function: utility, informativeness, and complexity.
>
>
> **Reusability (major)**
>
> To address the reviewer’s concern about the availability of our code, we made our code available by providing a link to an anonymized repository. Following the NeurIPS instructions we received, this link was posted in a separate comment to the AC. We included in the repository a demo notebook, useful for exploring the ManyNames dataset and identifying target and distractor objects in the images, which we hope will facilitate further extensions of our work.
>
>
> **Generalizability (minor)**
>
> We completely agree with the reviewer that our framework is general and could be used to study additional environments and open questions in the literature. We will happily add a short discussion on this in the paper.
>
>
> **Comments (minor)**
>
> Thanks for noting the typos, we will certainly fix them. We are also very excited about considering the Mandarin Chinese data in ManyNames in future work! We would have loved to do so in this paper, but given the scope of a single NeurIPS paper, we won’t be able to do justice to a cross-linguistic study in addition to our current set of results.

---

> > ### Comment · Reviewer_Cm9N · 2024-08-13
> >
> > Thank you for your rebuttal.  I believe that the proposed changes will sufficiently address my concerns about the framing and make the findings of the empirical clear both in the context of the paper and in the broader context of computational approaches to linguistics.  I will update the review's scores accordingly.

---

> ### Author Response · Authors · 2024-08-07
> **References**
>
> [1] Dowty, D. R., Wall, R. E., & Peters, S. (1981). *Introduction to Montague Semantics*. Springer.
>
> [2] Murphy, G. (2002). *The Big Book of Concepts*. The MIT Press.
>
> [3] Rosch, E. (1975). Family resemblances: Studies in the internal structure of categories. *Cognitive Psychology*, 7(4), 573-605.
>
> [4] Rosch, E., Mervis, C. B., Gray, W. D., Johnson, D. M., & Boyes-Braem, P. (1976). Basic objects in natural categories. *Cognitive Psychology*, 8(3), 382-439.
>
> [5] Regier, T., Kemp, C., & Kay, P. (2015). Word meanings across languages support efficient communication. In B. MacWhinney & W. O'Grady (Eds.), *The Handbook of Language Emergence* (pp. 237-263). Wiley-Blackwell.
>
> [6] Zaslavsky, N., Kemp, C., Regier, T., & Tishby, N. (2018). Efficient compression in color naming and its evolution. *Proceedings of the National Academy of Sciences*, 115(31), 7937-7942.
>
> [7] ​​Zaslavsky, N., Regier, T., Tishby, N., & Kemp, C. (2019). Semantic categories of artifacts and animals reflect efficient coding. *Proceedings of the Annual Meeting of the Cognitive Science Society*, 41.
>
> [8] Grice, Paul (1975). "Logic and conversation". In Cole, P.; Morgan, J. (eds.). Syntax and semantics. Vol. 3: Speech acts. New York: Academic Press. pp. 41–58.
>
> [9] Goodman, N., & Frank, M. (2016). Pragmatic language interpretation as probabilistic inference. *Trends in Cognitive Sciences*, 20.
>
> [10] Monroe, W., Hawkins, R. X. D., Goodman, N. D., & Potts, C. (2017). Colors in context: A pragmatic neural model for grounded language understanding. *Transactions of the Association for Computational Linguistics*, 5, 325–338
>
> [11] Franke, M. (2012). Signal to Act: Game Theory in Pragmatics. PhD thesis, University of Amsterdam.

---

### Official Review · Reviewer_zhL9 · 2024-07-13

**Soundness:** 2
**Presentation:** 3
**Contribution:** 2
**Rating:** 3
**Confidence:** 4

**Summary:**

This paper investigates emergent communication in artificial agents and how properties of the emergent language compares to properties of human language. Specifically, it aims to present a framework to study pragmatic language emergence in systems with different learning objectives, in order to determine constraints that lead to human-like linguistic systems. For training, they use a reference-game setup based in naturalistic images. Based on the closest similarity in complexity, lexicon size and lowest normalized information distance compared to the human baseline, they argue that agents need to optimize for all: utility, informativeness, and complexity.

**Strengths:**

- Well thought-through dataset preprocessing and task formulation
- Relevant and interesting topic for the computational linguistics and computation cognitive science community
- Thoughtful setup and analyses that are informative for understanding what the agents learn

**Weaknesses:**

- The paper overall argues for the necessity of a trade-off of different communicative pressures in order to learn a human-like emergent language -- as measured by the system's similarity to human language in complexity, lexicon size and Normalized Information Distance (NID). Based on these parameters, the authors argue that utility, informativeness, and complexity need to be jointly optimized for in order for a human-like language system to emerge. However, the best-performing system (according to the  complexity, lexicon size, and NID metrics) only achieves a final task performance of 72% -- If I understand the task setup correctly, the baseline chance performance is 50% and people should be expected to achieve nearly 100% in performance. The utility-only objective achieves an accuracy of 95% but does so with a larger lexicon size, worse NID and differently looking complexity. So why do the complexity/lexicon size/NID metrics hold so much more value than the actual task performance? **It seems to me that this argument can really just be made when all systems succeed on the task equally and conditioned on that, we can inspect which learning constraints appear to lead to the most human-like system.** And learning constraints that don't lead to task success while others do should already disqualify from further inspection. I find this methodologically problematic and to me this is a fundamental flaw in the argument. (It also severely lacks discussion in the paper.)
- Minor: The paper writing could generally be improved. Firstly, I find Figure 2 quite confusing as to what the labels mean and when something is masked. Secondly, there are various typos in the paper (e.g., lines 90, 153, 158, 170, 179) and confusing enumeration (line 125-129).

**Questions:**

- Could you elaborate on what naturalistic images afford compared to more controllable environments (e.g., more interesting lexicon size?)
- What is the basis for the claim that utility and informativeness are only partially aligned? (line 245)
- There is an argument brought forth that the agents are pressured to overcompensate because of the lack of syntax (lines 231-233). Could you elaborate on this a bit more? Why can syntax not emerge and wouldn't that then be a fundamental constraint on the comparability of the system to human language (especially when it comes to lexicon size as a metric)?
- In the utility-only condition, the agents learn to solve the task well, but the authors argue that the listener doesn't learn to reconstruct robust and non-contextual semantics. How is it possible that the model still generalizes so well?
- How do the pragmatic and semantic condition map onto the results? I understood the paper to say that the agents were trained in the pragmatic setting where it's the listener's goal to pick out the intended target based on two options. The utility evaluation seems to measure the listener's success to pick out the correct target. What is the semantic setup used for?

**Limitations:**

What questions does this leave unanswered? What simplifying assumptions are being made about language learning (no syntax or social learning)? The English language limitation is certainly true but not particularly meaningfully discussed -- why would we potentially expect variation across languages?

---

> ### Author Rebuttal · Authors · 2024-08-07
>
> **Weaknesses:**
>
> 1. Major: The reviewer assumed that “people should be expected to achieve nearly 100% in performance” and based on that, argued that we should disqualify systems that do not achieve near-perfect task utility. We would like to clarify that the reviewer’s claim is inconsistent with prior work showing empirical evidence that **people cannot achieve near-perfect performance in our task** (see additional comment for detail). This also makes sense intuitively: our study focuses on the lexicon, and therefore our task focuses on the use of **lexical items** (i.e., single words/signals). Humans cannot achieve near-perfect performance with lexical items, but rather need to employ syntactic constructions for that. This stems from the ambiguity of the human lexicon, an important property that is believed to facilitate efficient communication [e.g., 1; 2]. Therefore, we find it encouraging that our approach has identified a regime in which human-like behavior arises both at the lexical level — as captured by NID, lexicon size, and informational complexity — as well as at the pragmatic level, as captured by the (expectedly bounded) task performance. We briefly explained this in lines 226-233 in our submission, but thanks to the reviewer’s thoughtful comments we realize that we should elaborate more on this point and explain it more clearly. We will certainly do this for the final version.
>
>    Re the reviewer’s question on task performance vs other metrics: due to space limitation, we addressed this in the additional comment.
>
> 2. Minor: we will easily fix the typos and confusing enumeration. Furthermore, we will do a thorough English proofreading of the entire paper. The reviewer also noted that the labels in Fig2 are confusing, but they didn’t explain what is confusing about these labels (they seem rather intuitive to us). We therefore ask the reviewer for a clarification on this potential issue, to allow us to fully address all the concerns.
>
>
> **Questions:**
>
> 1. Our chosen dataset of naturalistic images has several advantages: (a) richer lexicons, as the reviewer suggested; (b) using images of naturalistic scenes is particularly crucial in our case, because the structure of the emergent communication is sensitive to the statistics of co-occurring objects in the environment; (c) controllable environments have been considered in prior work with the utility-complexity-informativeness tradeoff [3], we thus extend this framework to more naturalistic settings;
>
> 2. Optimizing utility and optimizing informativeness are only partially aligned because they are different objectives that can lead to different solutions, but at the same time, they don’t necessarily compete and can sometimes support each other. For example, high informativeness can facilitate high utility, as seen in Table 1 ($\lambda_I = 1$). The intuition for this is that if the listener can accurately reconstruct the speaker’s representation of the target object, then it could perform reasonably well in many downstream tasks. At the same time, optimizing utility alone can be achieved with very poor feature reconstruction (i.e., low informativeness), as can be seen in Table 1 ($\lambda_U = 1$) and Figure 4a. The intuition for this is that when there is no pressure for informativeness, the listener can learn arbitrary representation spaces (e.g., random projections) that don’t resemble what the speaker has in mind. While this may yield good task performance for in-distribution inputs, it’s unlikely to generalize well to out-of-distribution (see [4] for preliminary results).
>
> 3. As noted above, our work focuses on lexical systems, i.e., the meanings of single words, and not syntax. We therefore allow our speaker agents to send only one word/signal per interaction. Respectively, the English ManyNames data that we use is also based on single-word responses from people. Our measures of complexity — the lexicon size and the informational complexity term — are both measures of lexical complexity, not syntactic complexity. The reviewer is correct that syntax is also an important feature of language; it is beyond the scope of this work and we will  add a note on this matter in the limitation section.
>
> 4. This is best illustrated in Fig. 4a, which shows that in the utility-only condition the listener’s reconstructions are misaligned with the speaker’s representations of the target visual features, in contrast to other solutions (Fig. 4b and Fig. 4d) where the listener is better able to recover the speaker’s representations. We believe that the listener achieves high task performance in the utility-only condition because it learns random projections instead of useful reconstructions. The random representations could perform well for in-distribution inputs, but they are unlikely to generalize well to out-of-distribution inputs (see [4] for preliminary results).
>
> 5. The semantic setup is designed to recapitulate the human naming experiment with our artificial agents. In a naming experiment (such as the ManyNames experiment), participants are shown a single object and are asked to name it (using a single word). In contrast to the pragmatic task, there is no distractor and no downstream task. We use this setup to evaluate the emergent lexical systems with respect to the human naming data. NID, lexicon size, and complexity are all based on this setup.
>
> **Limitations:**
>
> Indeed, as the reviewer noted, this work doesn’t address the evolution of syntax. We focus on the evolution of the lexicon, which provides the building-blocks for syntax, and it is far from being well understood. We agree that this is a simplifying assumption and adding syntax is an important direction for future work. We will add a discussion on this in the limitations section.
>
> As for the question about linguistic variation: it is well documented in the literature that languages vary widely in the ways they structure the environment into words, see for example [5-7].

---

> ### Author Response · Authors · 2024-08-07
> **Further details and references**
>
> **Empirical evidence suggesting that humans do not achieve near-perfect performance:**
>
> First, recall that our work focuses on understanding the emergence of the lexicon, and therefore our task setup only considers the use of lexicalized items (i.e., single signals or words) in communication. This kind of experimental paradigm is very common in the cognitive science of language, and is often used for studying the emergence of the lexicon. Second, while we do not have human data for our specific task, prior work has collected behavioral data for closely related tasks in this domain:
>
> (a) The ManyNames dataset itself [9], which contains non-contextualized free naming data (i.e., with no distractor) from native English speakers, shows that the English lexicon contains unresolved ambiguities. This is not surprising given what we know about the lexicon and the probabilistic nature of many semantic categories [e.g., 1, 2, 5].
>
> (b) Mädebach et al. (2022) [8] conducted a human experiment in a pragmatic referential task similar to ours (with distractors), but in contrast to our setup, they did not restrict participants to single-word responses. Their findings suggest that English speakers often cannot use the lexicon alone to resolve visual ambiguities in the ManyNames images. For example, the lexicon does not allow speakers to unambiguously distinguish between two similar chairs in the same scene. Instead, in such cases participants would need to employ syntactic constructions that go beyond the lexicon itself (e.g., “the chair on the left”).
>
> While studying syntactic constructions is an important direction for future extensions of our work, our current paper focuses on how the semantic structure of the lexicon emerges, and this aspect of human language does not support perfect task performance.
>
>
> **On the importance of task performance vs other metrics for evaluation**
>
> > “The utility-only objective achieves an accuracy of 95% but does so with a larger lexicon size, worse NID and differently looking complexity. So why do the complexity/lexicon size/NID metrics hold so much more value than the actual task performance?”
>
> High task performance (utility) can be trivially achieved with very non-human-like communication systems. For example, by assigning a unique signal to each object in the dataset, or by using the same signal for entirely unrelated objects that never appear in the same context. Therefore, focusing only on task performance is unlikely to explain the emergence of human-like lexicons. Taken together with our previous point, that the English lexicon does not afford near-perfect task performance, we actually find it very encouraging that the emergent system that is most aligned with humans also achieves bounded task performance. In other words, we value all metrics for evaluation, including task-performance, and when comparing them with human behavior they all consistently support our conclusions.
>
>
> **References:**
>
> [1] Piantadosi, S.T., Tily, H.J., & Gibson, E. (2011). The communicative function of ambiguity in language. *Cognition*, 122, 280-291.
>
> [2] Zaslavsky, N., Kemp, C., Regier, T., & Tishby, N. (2018). Efficient compression in color naming and its evolution. *Proceedings of the National Academy of Sciences*, 115(31), 7937-7942.
>
> [3] Tucker, M., Levy, R., Shah, J., & Zaslavsky, N. (2022a). Trading off utility, informativeness, and complexity in emergent communication. *Advances in Neural Information Processing Systems*.
>
> [4] Tucker, M., Levy, R., Shah, J., & Zaslavsky, N. (2022b). Generalization and translatability in emergent communication via informational constraints. In *NeurIPS 2022 Workshop on Information-Theoretic Principles in Cognitive Systems*.
>
> [5] Malt, B. C. (1995). Category coherence in cross-cultural perspective. *Cognitive Psychology*, 29(2), 85–148.
>
> [6] ​​Zaslavsky, N., Regier, T., Tishby, N., & Kemp, C. (2019). Semantic categories of artifacts and animals reflect efficient coding. *Proceedings of the Annual Meeting of the Cognitive Science Society*, 41
>
> [7] He, Y., X. Liao, J. Liang, G. Boleda. 2023. The Impact of Familiarity on Naming Variation: A Study on Object Naming in Mandarin Chinese. *CoNLL 2023*.
>
> [8] Mädebach, A., Torubarova, E., Gualdoni, E., & Boleda, G. (2022). Effects of task and visual context on referring expressions using natural scenes. In *Proceedings of the Annual Meeting of the Cognitive Science Society*, 44.
>
> [9] Silberer, C., Zarrieß, S., and Boleda, G. (2020). Object naming in language and vision: A survey and a new dataset. In *Proceedings of the 12th Language Resources and Evaluation Conference*, pages 5792–5801, Marseille, France. European Language Resources Association.

---

> ### Comment · Reviewer_zhL9 · 2024-08-09
>
> Thank you for the details and clarifications. I have a few follow-up questions based on your response.
>
> Firstly, could you quantify the expected human performance on the task and how you infer that number? I'm assuming it's definitely higher than 50% because it's unlikely that there are more than two objects with the same lexical label (on average). Is it closer to 60%, 70%, 80%, 90%, 95%?
> (To get further insights on the performance, could you maybe leverage the data split that Mädebach et al. 2022 introduced into no-competitor, lexicon-sufficient, and syntax-needed? I think this might potentially lead to interesting observations on error patterns and whether the cases where the model fails might be cases where people would employ syntactic specification.)
>
> About the claim that *syntax is excluded from this analysis:* You specifically highlight that this work's contribution aims to only consider the semantics/pragmatics interface and tries to abstract away from syntax. I would like some clarifications on this point.
>  - Firstly, where do you draw the line between syntax and semantics in this particular setup? Mädebach et al. 2022 specifically discusses that they had to make some fairly ad-hoc decisions about what qualifies as lexical and syntactic modification. A prime example are compound nouns, where Mädebach et al. for instance chose a specific frequency threshold to determine whether they classify a noun compound as syntactic or lexical modification (making "tennis player" lexical specification and "front court player" syntactic specification). This is simply to illustrate that the syntax/semantic divide isn't as clear cut in this setting as you make it seem in your response and I would like some clarification on this point.
> - Secondly, human language has developed alongside syntactic complexity, so the English language has fundamentally one major additional dimension of system complexity that actively negotiates semantic complexity. As you pointed out in your response, in Mädebach et al's work, they find that people choose syntactic structures over lexical specification quite frequently. The option to have syntactic variation at a speaker's disposal for a reference game or reference disambiguation task can then be reasonably expected to decrease the lexical variation that will emerge in a learning system. In fact, there is ample of evidence in the cognitive science literature that different languages indeed optimize for a distinct syntactic complexity/semantic complexity trade-off (see, e.g., Reali et al., 2018 [1], for an interesting exploration of this trade-off). And even in an already established language, like English, recent work suggests that people trade off lexical and syntactic complexity against each other in production (Rezaii et al., 2022 [2]). Given this, how are you thinking about comparing a system that fundamentally has one more dimension for reducing referential ambiguity to a system that is aimed to be stripped from it?
> - This is a more minor point, but you highlight in your responses that you restricted the model learning setup to single words and that that prohibits learning of syntactic structures. Could you clarify how restricting an emerging language system to single words necessarily disables learning syntactic structure? Since the definition of what qualifies as a token is sort of arbitrary, couldn't the system in principle map to sophisticated morphosyntax? (To clarify, I don't believe that that's the case here but I would like clarification on how the single-word constraint relates to emerging-language syntax.)
>
> Other points:
> - You also stated that "High task performance (utility) can be trivially achieved with very non-human-like communication systems." -- I agree that utility is not a *sufficient* condition but what I raised in my review is the concern that a system that's not achieving task performance seems like it's missing a *necessary* condition to be an interpretable cognitive model.
> - Do you have estimates on how stable the learned systems are and how generalizable the results? I'm aware that the tested model is a ResNet model which has been shown in the past to have promising representational alignment with humans. However, there are still many ad-hoc decisions that are part of training and evaluating such models as cognitive models which might significantly affect the results. For instance, is there evidence that the stopping criterion used for training is a reasonable stopping criterion for finding human alignment? If you were to continue training would the learned system significantly change?
>
> Citations:
>
> [1] Reali, Chater, Christiansen (2018), "Simpler grammar, larger vocabulary: How population size affects language", Proceedings of the Royal Society B
>
> [2] Rezaii, Mahowald, Ryskin, Dickerson, Gibson (2022), "A syntax–lexicon trade-off in language production", PNAS

---

> ### Author Response · Authors · 2024-08-13
> **Responses to follow-up questions - Part 1**
>
> We thank the reviewer for engaging in an interesting discussion about our work. Before turning to our detailed response (below), we would like to highlight that none of the points that the reviewer raised seem to be grounds for rejection. We have already addressed the concerns that were raised in the review, and the reviewer’s main follow-up questions about the syntax-semantics interface are not specific to our work. These questions touch upon some of the deepest open challenges in cognitive science and linguistics. Our work does not attempt to address these challenges. Instead, we follow common practices in the field and make well-established simplifying assumptions in order to advance the understanding of how lexical systems may emerge from contextual pragmatic interactions. Furthermore, the reviewer’s first follow-up question, on estimating the expected human performance, has actually helped us to further strengthen the support for our results and conclusions. We therefore hope that the reviewer will positively reconsider the rating of our paper.
>
> Detailed response to the reviewer's questions:
>
> > Firstly, could you quantify the expected human performance on the task and how you infer that number?
>
> While the Mädebach et al. data does not allow us to quantify precisely the human performance in our task (our point in the paper, lines 226-233, and in our rebuttal was qualitative rather than quantitative), we can use that data to roughly estimate an upper and lower bound on the human performance. For the upper bound, notice that Figure 2 in Mädebach et al. shows that, across all context conditions (no-competitor, lexicon-sufficient, and syntax-necessary), the performance is around 80%, and recall that Mädebach et al. did not restrict the participants’ responses to lexical items. Therefore, it is unlikely that in our task, which is restricted to lexical items, the performance would be better than 80%. For the lower bound, notice that in the most favorable condition, i.e., the lexicon sufficient condition, which could in principle be solved using the lexical items, the proportion of responses that correspond to lexical items without syntactic construction is only ~42% (‘no specification’ and ‘lexical’ response type). In ~39% of the response participants use syntactic construction even though there is a lexical item that could unambiguously describe the target. One possible explanation for this behavior is that in some cases lexical retrieval is harder than syntactic constructions (e.g., for infrequent words). Therefore, while we expect the proportion of lexical responses to be higher than ~42% in our version of the task, we also expect the error rate to be higher. While we don’t have a way to quantify this further, we do agree with the reviewer that the task performance would probably be higher than 50%.
> To summarize: the Mädebach et al. data suggests that the human performance in our task would be somewhere between 50%-80%. In comparison, our model achieves ~72%, which is well within the reasonable range of human performance. We find this estimation very helpful to further support our results and conclusions, and thank the reviewer for suggesting it!
>
> > where do you draw the line between syntax and semantics in this particular setup?
>
> Syntax corresponds to how single words can be combined into larger linguistic structures, such as phrases or sentences. Lexical semantics corresponds to word meanings. These two subfields of linguistics are largely separable, but indeed, as the reviewer notes, there are gray areas in between. However, these gray areas are irrelevant to our work because our agents simply cannot learn them. Recall that in our setup communication signals cannot be combined and in each communication act only a single signal can be generated. Therefore borderline cases like the noun compound, which require combining words, cannot emerge in our agents. Instead of lexicalized compounds, our agents would simply use a single communication signal, thus bypassing the issue that Mädebach et al. had to address in the human data.

---

> > ### Author Response · Authors · 2024-08-13
> > **Responses to follow-up questions - Part 2**
> >
> > > how are you thinking about comparing a system that fundamentally has one more dimension for reducing referential ambiguity to a system that is aimed to be stripped from it?
> >
> > First, we would like to emphasize that we are comparing the agents’ lexicon with the human lexicon. That is, the reviewer's claim is factually inaccurate in the sense that our analysis does not include the additional syntactic dimension of English, but rather focuses only on the lexical dimension of English.
> >
> > Second, while it is certainly true that the human lexicon evolved together with syntax, in contrast to the emergent lexicon in our agents, this simplifying assumption is very common in the emergent communication literature, and more generally, in agent-based and game-theoretic approaches to language evolution. Furthermore, this kind of simplifying assumption – i.e., studying only one key aspect of human cognition, even though it has evolved together with many other cognitive functions – applies to every cognitive model that we are aware of and seems inevitable, at least given the current state of the field.
> >
> > Finally, we would like to highlight that we have already acknowledged in the rebuttal that this simplifying assumption should be addressed more explicitly in the limitations sections of the paper and we certainly intend to do so if given the opportunity to revise the paper.
> >
> > > Could you clarify how restricting an emerging language system to single words necessarily disables learning syntactic structure?
> >
> > In our model, agents learn a codebook of k communication vectors, similar conceptually to the idea of word embeddings. As explained above, at each communication round the speaker can generate only a single communication vector, which rules out the possibility of syntactic structures that emerge from combining communication signals. As for the possibility of a complex morphological structure in the vector embeddings, this is unlikely to emerge because the agents cannot reuse subparts of the communication vectors, but rather only the vectors as a whole. To address this, Tucker et al. (2022) proposed an extension of the model that does support some degree of combinatorial structure within the communication signals, but we have not used that extension in our work.
> >
> > > You also stated that "High task performance (utility) can be trivially achieved with very non-human-like communication systems." -- I agree that utility is not a sufficient condition but what I raised in my review is the concern that a system that's not achieving task performance seems like it's missing a necessary condition to be an interpretable cognitive model.
> >
> > Thank you for clarifying your point. We hope that our rebuttal and response to your first follow-up question here have convinced you that this is not a necessary condition. Human lexical systems do not seem to afford near-perfect task performance, and therefore it would not make sense to consider only emergent systems that achieves near-perfect task performance.
> >
> > > Do you have estimates on how stable the learned systems are and how generalizable the results? I'm aware that the tested model is a ResNet model which has been shown in the past to have promising representational alignment with humans. However, there are still many ad-hoc decisions that are part of training and evaluating such models as cognitive models which might significantly affect the results. For instance, is there evidence that the stopping criterion used for training is a reasonable stopping criterion for finding human alignment? If you were to continue training would the learned system significantly change?
> >
> > We have verified that our results are robust across random seeds (see Table 1). The stopping criteria we used is convergence of the training objective, which is one of the most common, generic stopping criteria in the literature. We have not considered a stopping criteria that specifically targets better alignment with humans, but that presumably could only improve our results. For other hyperparameters, we have built on the prior work by Tucker et al. (2022) that established the VQ-VIB framework.

---

> > > ### Comment · Reviewer_zhL9 · 2024-08-13
> > >
> > > I thank the authors for their detailed responses. I am going to stick with my current score.

---

### Author Rebuttal · Authors · 2024-08-07

We thank all the reviewers for their helpful and thoughtful comments. We are excited that the reviews are generally positive and in favor of publication. As we explain in our detailed response to each reviewer and in the summary below, we believe that all the concerns raised by the reviewers can be easily addressed with minor revisions to the paper that will clarify a few points about the framing and significance of our work.

Below is a summary of our responses to the key points raised in the reviews, as well as the revisions we intend to implement to address all the reviewers’ concerns.

**1. zhL9’s major concern**

We believe that the major concern raised by zhL9, which seems to be the grounds for this reviewer’s negative rating, is based on a misinterpretation of our work and does not take into account prior work on how humans would perform in our task. Specifically, the reviewer assumed that humans perform near-perfectly in our task, without providing any references to support that claim, and based on that, argued that we should have excluded systems that do not solve the discrimination task perfectly. However, as we explain in our detailed response to zhL9, the reviewer’s assumption is unsupported by prior work that provides empirical evidence in direct support of our findings. That is, humans cannot solve the task perfectly, similar to our model that is best aligned with the English lexicon. Therefore, the fact that we are able to identify, in a principled way, a regime in which human-like behavior emerges both at the lexical semantics level — as captured by the resemblance between the emergent lexicon and the English lexicon in our domain — and at the pragmatics level — as captured by the bounded task performance — is not a design flaw but rather a major contribution of our work.

We suspect that this confusion may have stemmed from misinterpreting our work as focusing on language as a whole, including syntax, rather than focusing specifically on the semantics-pragmatics interface as highlighted throughout the paper (including in the title and abstract).

We hope that we addressed zhL9’s major concern, both here and in our detailed response to the reviewer. We greatly appreciate zhL9’s comments and feel that clarifying this point would improve the presentation of the paper. Specifically, we intend to address this issue by:

(a) Further elaborating on the empirical evidence suggesting that humans do not solve our task perfectly.

(b) Clarifying that achieving bounded task performance is a desired (human-like) property for lexical systems (this is not the case for syntax, but that is not within the scope of our work).

(c) Clarify that our work does not consider syntax, and discuss the important extension to syntax in the limitations section of the paper.


**2. Clarity of framing**

Reviewers Cm9N and F8cg raised related concerns about the framing of the paper. Specifically, Cm9N noted that while the title and introduction capture the correct framing, other parts of the paper seem to reflect a somewhat different framing. As explained in our response, we believe that the abstract, conclusions, and related work sections also capture the correct framing, but we agree with the reviewer that the framing in the more technical parts of the paper could be improved. The reviewer also offered a constructive suggestion for addressing this issue, by elaborating on what semantics and pragmatics are, why their distrintinction is important, and how our work is significant in this context. We are grateful for this suggestion and will address it along the lines of our detailed response to Cm9N, as follows:

(a) Extend the first paragraph of the introduction to clarify what we mean by semantics and pragmatics.

(b) Adjust the related work section to clarify the distinction between semantics and pragmatics, in addition to the significance of bridging them.

(c) Revise the technical sections to ensure that they are also contextualized w.r.t. the correct framing of the paper.

Relatedly, F8cg wondered about the significance of training in a pragmatic setting and testing in a semantic setting. We believe that the revisions described above will address this concern as well, as we explain in our detailed response to F8cg.

**3. Differentiation from prior work**

F8cg raised a concern that our work uses the same communication model from Tucker et al. (“Generalization and Translatability in Emergent Communication via Informational Constraints.”). As we explain in our detailed response to F8cg, this is factually inaccurate and there are three key differences between our model and the Tucker et al model. Most importantly, our new setting is designed to address a major open question at the interface between semantics and pragmatics, which Tucker et al. have not considered and their communication model would not support. Having said that, we agree with the reviewer that these key differences w.r.t. Tucker et al. were not conveyed clearly enough in the paper. To address this issue, we will clarify this point in the related work section, around lines 113-117, where we discuss the prior work by Tucker et al.


**4. Other questions and concerns**

- Cm9N’s reusability concern: to fully address this concern, we have provided an anonymized link to our code (in a comment to the AC, following the NeurIPS instructions).


- We will fix all the typos and will do a thorough English proofreading of the entire manuscript.

- The reviewers also raised several valuable questions. We intend to incorporate our clarifications in response to these questions in the revised version of the paper.

---

### Decision · Program_Chairs · 2024-09-25

**Decision:**

Accept (poster)

**Comment:**

This paper on emergent communication systems explores the co-evolution of pragmatics and lexical semantics. While it presents interesting methodology and results, it is somewhat borderline. The most expert reviewer raised significant concerns about the trade-off between pragmatic objectives and task performance, questioning the validity of the main argument. Despite this concern, which I assess the authors have responded satisfactorily to and to which the reviewer did not completely respond to, and given the paper's potential contributions to the field and the positive aspects noted by this reviewer and by other reviewers, I recommend acceptance. The authors should address the raised issues in the camera-ready version to strengthen the paper.